# Molecular Mechanisms Underlying Intensive Care Unit-Acquired Weakness and Sarcopenia

**DOI:** 10.3390/ijms23158396

**Published:** 2022-07-29

**Authors:** Marcela Kanova, Pavel Kohout

**Affiliations:** 1Department of Anaesthesiology and Intensive Care Medicine, University Hospital Ostrava, 708 52 Ostrava, Czech Republic; 2Institute of Physiology and Pathophysiology, Faculty of Medicine, University of Ostrava, 703 00 Ostrava, Czech Republic; 3Department of Internal Medicine, 3rd Faculty of Medicine, Charles University Prague and Teaching Thomayer Hospital, 140 59 Prague, Czech Republic; kohoupav@seznam.cz

**Keywords:** intensive care unit-acquired weakness, sarcopenia, proteostasis, ubiquitin–proteasome system, rapamycin system, muscle atrophy

## Abstract

Skeletal muscle is a highly adaptable organ, and its amount declines under catabolic conditions such as critical illness. Aging is accompanied by a gradual loss of muscle, especially when physical activity decreases. Intensive care unit-acquired weakness is a common and highly serious neuromuscular complication in critically ill patients. It is a consequence of critical illness and is characterized by a systemic inflammatory response, leading to metabolic stress, that causes the development of multiple organ dysfunction. Muscle dysfunction is an important component of this syndrome, and the degree of catabolism corresponds to the severity of the condition. The population of critically ill is aging; thus, we face another negative effect—sarcopenia—the age-related decline of skeletal muscle mass and function. Low-grade inflammation gradually accumulates over time, inhibits proteosynthesis, worsens anabolic resistance, and increases insulin resistance. The cumulative consequence is a gradual decline in muscle recovery and muscle mass. The clinical manifestation for both of the above conditions is skeletal muscle weakness, with macromolecular damage, and a common mechanism—mitochondrial dysfunction. In this review, we compare the molecular mechanisms underlying the two types of muscle atrophy, and address questions regarding possible shared molecular mechanisms, and whether critical illness accelerates the aging process.

## 1. Introduction

Intensive care unit-acquired weakness (ICUAW) brings about skeletal muscle wasting due to critical illness and has important clinical implications, significantly impacting rehabilitation, and increasing both morbidity and mortality. ICUAW is sometimes referred to as critical illness polyneuromyopathy—being called critical illness polyneuropathy (CIP) when nerve involvement predominates, or critical illness myopathy (CIM) where muscle involvement is crucial. It manifests as muscle weakness that develops rapidly, prior to any detectable muscle wasting. We typically see symmetrical limb weakness, which is more pronounced in the proximal limbs (shoulders, hips). The diaphragm and intercostal muscles are also affected, with associated difficulties in discontinuing artificial lung ventilation (failure to withdraw mechanical ventilation) and long-term disability, known as ventilator-induced diaphragmatic dysfunction [1,2].

It has an incidence ranging from 25–31% [3], with the extent depending on the disease itself, but also on patient treatment. Muscle atrophy is aggravated by glucocorticoid therapy, prolonged analgosedation, neuromuscular blocking agents, immobilization, and artificial pulmonary ventilation. These are typical procedures for patients with ARDS, including patients with COVID-19. In patients with respiratory insufficiency on artificial lung ventilation, severe ICUAW is found in 25–75% of patients [4]. Inflammatory factors can cause axonal swelling in motor neurons, resulting in “denervation” or neurapraxia. Rapid muscle loss occurs in septic states in response to microbial invasion (PAMPs, pathogen-associated molecular pathways) or in synergy with alarmins released from damaged organs (DAMPs, damage-associated molecular pathways), which leads to the activation of stress metabolism. Muscle loss is further accelerated by metabolic acidosis and insulin resistance. In cases of severe injury, mitochondrial damage is significant, inducing progressive persistent inflammation and catabolism (Persistent inflammation, Immunosuppression, and Catabolism syndrome or PICS), and subsequently, muscle loss [5].

Sarcopenia is age-related skeletal muscle decline. Sarcopenia is characterized by low levels of response to three parameters: (1) muscle strength, (2) muscle quantity/quality, and (3) physical performance—an indicator of severity. The loss of muscle mass (cross-sectional area) and strength begins to manifest after the age of 55 due to an imbalance between protein synthesis and protein breakdown. Its occurrence is highly variable, depending on many factors, of which endurance training, sedentary lifestyle, healthy diet, and protein intake seem to be crucial. From approximately 50 years of age, muscle volume begins to decline by about 1% per year, or 8% per decade, and muscle strength declines by 1–5% per year. Elderly patients with already expressed sarcopenia are among those with the highest risk of developing ICUAW [6,7].

In addition to enabling movement and maintaining postural tone, muscles also have an endocrine role. The muscle cell produces myokines during contraction, which act on virtually all organs, from the CNS (its metabolism of kynurenine with reduction of depression has a neuroprotective effect), through affecting immune function by suppressing inflammation, to having important roles in protein and sugar metabolism, acting on the liver, pancreas, intestine, and improving insulin sensitivity. Thus, muscle loss has severe long-term consequences [8].

ICUAW is fundamentally a clinical state, and it was our aim in this review to emphasize that aspect, especially from the point of view of clinicians and intensive care specialists. Upon recovering from critical illness, often including organ support (such as long-term ventilation support, renal replacement therapy, or ECMO), patients often suffer long-term adverse consequences—mainly polyneuromyopathy, with cognitive and mental impairment—which significantly decreases quality of life.

## 2. Proteostasis

In healthy muscles, there is a constant turnover of muscle proteins, and a balance between proteosynthesis and proteolysis is essential for maintaining muscle mass. These processes alternate according to anabolic stimulation (diet, exercise). Proteosynthesis occurs in the postprandial phase, whereas proteolysis occurs in the postabsorptive inter-meal period. This ability of muscle mass to constantly renew itself allows it to respond to the rapidly changing needs of the organism. Once muscle glycogen is depleted, the muscle releases amino acids as an energy source for gluconeogenesis. Recovery of muscle fibers following exertion allows stimulation of proteosynthesis by dietary protein (leucine, hydroxymethylbutyrate) and peptides from damaged fibers (myofibrils) [9,10]. 

With advancing age, anabolic stimulation is rendered “dull”, and proteosynthesis is no longer capable of reaching the necessary turnover, and higher doses of protein are needed to stimulate postprandial proteosynthesis. This advancing anabolic resistance is due to insulin resistance, poorer blood supply to muscles with reduced nutritive flow, reduced endothelial function, etc. Proteolysis is largely unchanged, gradually leading to loss of muscle mass, muscle atrophy, and sarcopenia [11] (Figure 1).

The balance between proteosynthesis and proteolysis is shifted in favor of catabolism in critically ill patients due to a variety of causes such as sepsis, SIRS, immobilization, etc., and following a rapid decline in muscle strength, leads to muscle atrophy, and the development of ICUAW. Several pathophysiological changes occur as the body attempts to survive as part of the body’s metabolic response to stress (sepsis, acute illness, trauma, surgery): activation of the sympathetic nervous system, the release of catabolic hormones (catecholamines, glucocorticoids) and pro-inflammatory cytokines. The resulting anabolic resistance then inhibits proteosynthesis, and massive proteolysis is activated [12] (Figure 1).

### 2.1. Stress Metabolism

These are several mechanisms, well proven over the course of evolution, allowing the rapid restoration of organismal homeostasis. Sir David Cuthbertson divided this stress response into two phases. In the initial ebb phase, there is a transient drop in energy turnover for a few hours. This is followed by a significantly longer hypermetabolic flow phase when energy turnover increases. The sympathetic nervous system reacts first, releasing noradrenaline and adrenaline, allowing control over internal organs. Subsequently, the hypothalamic–pituitary–adrenal axis is activated. Catabolic hormones (catecholamines, cortisol, glucagon), counteract the anabolic insulin and growth hormone. This supplies the body with sufficient energy in the form of glucose. The degree of hyperglycemia in a critically ill patient reflects the degree of catabolism: the severity of the clinical condition. Glycogenolysis and subsequent gluconeogenesis in the liver ensure glucose supply, which is additionally redirected to vital organs by peripheral insulin resistance. Unlike peripheral tissues and muscles, the heart and brain are not dependent on insulin (GLUT1,3 transporters). Insulin resistance results from defects in the post-receptor insulin signaling pathways and from the downregulation of glucose transporters (GLUT4) in skeletal muscle and fat tissue. Muscle is a source of amino acids for gluconeogenesis. Fat utilization is suppressed in the acute phase by catabolic hormones. Over time, mitochondrial DNA is damaged by pro-inflammatory cytokines (TNF, IL1, IL6) and an increase in oxidative stress (reactive oxygen species, ROS). The extent to which mitochondrial function (the Krebs cycle, aerobic metabolism, and ATP production) is impaired will affect the prognosis of the patient [12,13,14].

### 2.2. Upregulation of Muscle Protein Breakdown

The stress metabolism described above is an evolutionarily validated mechanism to survive critical disease. However, the organism pays a “tax” in the form of muscle loss. The upset balance, uncontrolled degradation of muscle protein, and massive losses of myosin and other proteins (especially branched-chain amino acids, BCAA) are consequences of the activation of four main proteolytic systems: the ubiquitin–proteasome system (UPS), calpain, caspase, and the autophagy–lysosomal system [3,13,14].

#### 2.2.1. The Ubiquitin–Proteasome System (UPS)

UPS is the major proteolytic mechanism involved in the degradation of most damaged proteins, which are tagged (ubiquitination) and subsequently degraded in the 26 S proteasome. The process involves three steps regulated by interlinked ATP-dependent enzymatic reactions that finally results in ubiquitin being attached to the protein to be degraded. 

The E1 ubiquitin-activating enzyme acts as an “alarm clock”, activating ubiquitin. The ubiquitin-conjugating E2 enzyme then binds this activated ubiquitin. The final step is the action of an E3 ligase—in muscle, these are MuRF1 (muscle-specific RING finger protein1) and MAFbx (atrogin-1). E3 covalently attaches ubiquitin to lysine residues of the target substrate. The target protein is tagged with one or many (either several single or interlinked chains) ubiquitin molecules. The ubiquitinated protein is subsequently recognized and degraded by the 26 S proteasome into individual peptides. The 26 S proteasome is a large 2.5 MDa macromolecule equipped with several proteases for protein degradation. It consists of several subunits, the 20 S core protease (CP) with peptidase activities, capped at one or both ends by the 19 S regulatory particle (RP). The CP can cleave a broad range of polypeptides. The UPS system is designed to degrade short-lived regulatory polypeptides or damaged polypeptides. Although UPS plays a major role in degrading skeletal proteins and in muscle atrophy, it cannot release actin and myosin from the cytoskeleton. That requires the activity of two other prominent proteolytic systems, calpain and caspase3 [13,15,16]. 

Sepsis activates the proteasome system, with inflammation increasing ubiquitination and the degradation of inhibitors of NFκB (IkB). TNF and NFκB activate both muscle-specific E3 ligases (MuRF1 and atrogin-1), and consequent muscle proteolysis in both respiratory and limb muscle (increases of 30% and 50%, respectively, compared to controls [17]). Respiratory muscles are often affected—especially the diaphragm due to sepsis and inactivity—and this makes withdrawal from the ventilator difficult due to ventilator-induced diaphragmatic dysfunction (VIDD [17]).

#### 2.2.2. Calpains and Caspases

Calpains are calcium-dependent cysteine proteases, and the family is made up of 14 members; calpain μ and calpain m are pervasively expressed in muscle atrophy. There is a muscle-specific calpain 3, known as p94, that is also involved in some muscular dystrophies. Calpain activity is increased in sepsis, and they are involved in myosin cleavage [13]. 

Caspases also have the same function; they are also associated with cell death and apoptosis (caspase3) in addition to muscle atrophy [3,13].

#### 2.2.3. Autophagy–Lysosomal System

Autophagy is a basic pathway of skeletal muscle catabolism, being capable of degrading larger cellular structures. It is a necessary pathway for removing affected proteins and non-functional organelles (mitochondria—mitophagy, and proteasome—proteophagy). On the one hand, autophagy is necessary to ensure “healthy” muscle, but under a variety of catabolic conditions, and under the effect of pro-inflammatory cytokines, oxidative stress, or fasting its “purging” effect, is overactivated. Autophagy is regulated by FoXO (forkhead box-O class), mTORC1, LC3, and Atg7 (more on this below in Section 4.3). In such conditions, the autophagy–lysosomal system becomes dysregulated, leading to excessive degradation of muscle proteins. Along with an activated UPS, this aggravates muscle atrophy and the consequent imbalance in proteostasis leads to ICUAW in the critically ill [3,18]. 

### 2.3. Molecular Mechanisms Activating Muscle Protein Degradation

Upregulation of proteolysis is typical for ICUAW. Of the four systems mentioned, UPS is primarily involved in muscle catabolism. It leads to rapid degradation of actin and myosin and is an important source of amino acids for gluconeogenesis.

The TNF/NFκB/UPS axis is activated, leading to protein breakdown. The same axis is induced also by myostatin and glucocorticoids and the stimulation of proteosynthesis via the inhibition of the IGF-1/Akt/mTOR (mechanistic target of rapamycin) axis is inhibited. On the one hand, Akt activates proteosynthesis by activating mTOR, and on the other hand it leads to FoXO phosphorylation (thereby inactivating it) and the inhibition of proteolysis. Inactivating the IGF-1/insulin/Akt axis also inhibits downstream phosphorylation of FoXO. FoXO is thus retained in its active form and contributes to the stimulation of protein breakdown via the UPS (Figure 2) [3,19].

Ubiquitin, a member of the heat-shock protein family, is found in all cells and is activated during all ATP-requiring reactions. However, it is unable to cleave large myofibril molecules. This initial proteolytic step is done by caspase3 that cleaves actomyosin/myofibrils to produce substrates for the 26 S proteasome (UPS). The activation of caspases and subsequent proteolysis is precisely coordinated with the activity of the proteosynthetic mTOR system. The activation of one inhibits the activity of the opposing system. The inhibition of proteosynthesis is an adaptation to starvation and nutrient deprivation. mTOR inhibition activates overall protein degradation by the UPS as well as by autophagy. In nutrient-rich environments, mTOR decreases protein degradation by the UPS and stabilizes long-lived proteins, but it does not affect the breakdown of short-lived (misfolded or damaged) proteins [20].

#### 2.3.1. The PI3K/Akt Signaling Pathway

Which cellular signals activate muscle protein degradation? Identifying the initial step in UPS activation may help in therapeutically managing the rapid progression of muscle atrophy in some catabolic diseases, including ICUAW or age-related sarcopenia.

The inhibition of phosphatidylinositol 3 kinase (PI3K) activity and Akt kinase (which inhibits mTOR and proteosynthesis) is the signal for UPS activation and initiation of proteolysis in all cells in a catabolic state (sepsis, inflammation, immobility). Downregulation of Akt activity unblocks FoXO, and this transcription factor upregulates the expression of a critical E3 ubiquitin conjugation enzyme: atrogin-1/MAFbx. Low PI3-kinase activity induces the proapoptotic factor BAX, which then releases cytochrome c from mitochondria into the cytoplasm, which in turn increases caspase3 activity (Figure 2) [13,20]. 

MAFbx/atrogin-1 and MuRF-1 are two muscle-specific E3 ubiquitin ligases. MuRF-1 targets thick myosin filaments, thus triggering the predominant loss of red muscle fibers during ICUAW [11]. Unlike mTOR inhibition, which enhances proteolysis within 30 min, FoXO-mediated protein degradation requires hours for the transcription and translation of ubiquitin ligases to take effect [16,21].

#### 2.3.2. The Glucocorticoid Pathway

Glucocorticoids (GC) are known to potentiate muscle wasting in sepsis. The GC pathway is the second signaling cascade in proteolysis, activating both major proteolytic cascades—caspase3 and UPS. Nuclear factor kappa B (NFκB) and its inhibitor (IkB) oppose each other in UPS activation, and pro-inflammatory cytokines lead to the degradation of IkB in sepsis. NFκB then increases MuRF-1 expression, suggesting a role in muscle-specific atrophy. This increase can be inhibited by glucocorticoid receptor antagonists. The glucocorticoid receptor also enhances signaling of another atrophic factor myostatin, a member of the transforming growth factor (TGF-ß) family [22,23].

### 2.4. Downregulation of Muscle Protein Synthesis and Its Role in ICUAW and Especially in Sarcopenia

The reduction of protein synthesis, which starts almost immediately under the influence of pro-inflammatory cytokines, contributes to ICUAW, while immobilization and denervation contribute more gradually, and the rate of protein breakdown exceeds that of protein synthesis. The situation is different in sarcopenia, where anabolic resistance and reduction of protein synthesis are the main factors in the development of muscle atrophy. One of the causes of muscle anabolic resistance—where muscles do not respond adequately to stimulation, mainly by amino acids (leucine, hydroxymethybutyrate)—is impaired nutritive blood flow, with not enough anabolic products reaching their destination due to impaired microcirculation [13,14].

#### 2.4.1. Insulin/IGF1 Signaling

The main anabolic signaling cascade is the pathway from insulin/IGF-1 (insulin-like growth factor) to Akt (a serine/threonine kinase), which is activated upon energy availability. IGF1 is a tyrosine kinase receptor, and recruits PI3K, subsequently increasing the level of phosphatidylinositol-3,4,5-triphosphate (PtdIns(3,4,5)P3), which activates Akt.

Akt has multiple effects, the main one being activation of the mTOR system, which leads to increased proteosynthesis. On the other hand, Akt also phosphorylates the FoXO transcription factor and inactivates it by preventing its nuclear translocation. It subsequently inhibits the expression of E3 ubiquitin ligases (MuRF-1 and atrogin-1), thus inhibiting proteolysis. In addition, Akt phosphorylates a variety of downstream effectors which modulate the glucose transporter type 4 (GLUT4) and, thus, glucose uptake and glycogen synthesis by glycogen synthase kinase 3 (GSK3) [13,14].

Apart from the Akt signaling pathway, IGF-1 also activates the mitogen-activated protein kinase (MAPK) cascade, increasing myoblast proliferation. Sepsis and disuse of muscle reduce IGF-1 activity, via reduced mTOR activity, which results in decreased muscle protein synthesis [13,24].

#### 2.4.2. mTOR Signaling

mTOR is the main anabolic regulator ensuring cell growth and proliferation under anabolic conditions, following stimulation by growth factors, insulin, and nutrients (especially sufficient amino acids). One of its important functions is to maintain the available amino acid pool by regulating protein translation through effector molecules such as ribosomal protein S6 kinase (S6K1), which regulates protein translation initiation and E4-binding protein 1 (E4-BP1). The latter acts as an inhibitor of protein translation initiation by binding to and limiting the activity of eukaryotic initiation factor 4E (eIF4E). mTOR phosphorylates E4-BP1, thereby releasing eIF4E and allowing ribosomal cap-dependent translation to begin. mTOR is also involved in the regulation/activation of lipid synthesis [25,26]. 

Similar to Akt, mTOR is a serine/threonine kinase consisting of 2549 amino acids and several conserved domains. The FKBP12–rapamycin-binding (FRB) domain is the site of inhibitory action of the FKBP12–rapamycin complex. Other domains in mTOR are the kinase domain (KD), repressor domain (RD), and functionally the most relevant, the FAT carboxy-terminal domain (FATC) regulating mTOR kinase activity. mTOR binds other proteins to form intricate multiprotein complexes [25]. 

The first such protein is mTORC1, which ensures protein synthesis by initiating translation following the conjugation of eukaryotic initiation factor 4E (eIF4E) to the ribosome cap. It has a central role in the regulation of energy homeostasis. mTORC2, apart from influencing energy metabolism, is involved in the organization of the actin cytoskeleton and is an important activator of muscle growth. The main activating signaling pathway is the IGF/PI3K/Akt/mTOR pathway (Figure 3).

The structures of mTORC1 and mTORC2 complexes are highly similar, with both consisting of Deptor (death domain containing mTOR-interacting protein) and mLST8 (G protein βsubunit-like protein) complexes. They differ in the important inhibitory site for rapamycin: while mTORC1 contains raptor (regulatory-associated protein mTOR) and PRAS40 (proline-rich Akt/PKB substrate 40kDA), mTORC2 contains rictor (rapamycin-insensitive companion mTOR) and stress-activated protein kinase (mSin1) [25,26]. 

Stress, nutrient deprivation, and rapamycin are able to block the activity of mTORC1—i.e., proteosynthesis—and simultaneously activate proteolysis. mTOR inhibition activates overall protein degradation by UPS as well as by autophagy [21,22]. The two systems have opposing activities—mTOR activation inhibits autophagy, while lack of energy and amino acids inhibits mTOR and stimulates phagophore formation, as mTOR interacts directly with the ULK1 kinase complex and with death-associated protein1 (DAP1) [21,25]. New findings on nutrient-sensing regulatory mechanisms of mTORC1 activity reveal a more complex system, with amino-acid-dependent activation of mTORC1 (leucine, arginine, glutamine) demonstrating particular potency, and glucose-dependent regulation of mTORC1 also playing an important role. Under glucose-rich conditions, mTORC1 activation is induced by insulin. On the other hand, ATP deficiency and glucose deprivation activate the adenosine monophosphate kinase (AMPK), which decreases mTORC1 activity and induces autophagy. Taken together, the glycolytic flux regulates mTORC1 activity and mTORC1 activation enhances glycolysis to support cell growth under nutrient-rich conditions. Another possibility is oxygen-dependent regulation where hypoxia-inducible factor 1 (HIF-1) inhibits mTORC1 and activates autophagy. Reactive oxygen species (ROS) play a similar role [26]. 

Dysregulation of PI3K/Akt/mTOR leads to a decrease in proteosynthesis and contributes to the development of muscle atrophy and sarcopenia. This disruption of a key role in the regulation of cell growth and proliferation is also implicated in a variety of other diseases, including carcinogenesis and neurodegenerative diseases [21,25].

### 2.5. Disruption of Protein Turnover

Continuous protein turnover is affected in acute disease with the predominant activation of proteolysis and is especially prevalent in ICUAW. In contrast to this, anabolic sensitivity and, therefore, the ratio of proteosynthesis to muscle recovery decreases with age, resulting in the progressive development of sarcopenia.

## 3. Early Muscle Weakness in ICUAW

Muscle weakness manifests very early, especially in sepsis patients with the development of ICUAW. This weakness is seen long before we are able to detect any loss of muscle mass (cross-sectional area), or loss of muscle protein through inflammation-mediated proteolysis. Mechanisms contributing to muscle weakness include impaired intracellular Ca^2+^ homeostasis resulting in reduced fiber contractility, mitochondrial dysfunction with bioenergetic failure, channel dysfunction with membrane inexcitability, and hyperglycaemic toxicity [1,4].

### 3.1. Caspases, Calpains, and the Disruption of Myofilament Structure

Functionally, there are two types of muscle fibers. Slow-twitch muscle fibers, known as type I or red fibers (high blood supply) contain more mitochondria and myoglobin. They are aerobic, fatigue-resistant, and are focused on postural control. Fast-twitch muscle fibers—type II or white, anaerobic fibers provide a more powerful force, but over a shorter duration and fatigue quickly. These type II fibers are more sensitive to the multifactorial injury caused by critical illness. 

The integrity of contractile myofilaments and the regular striated pattern of actin and myosin are essential for sufficient muscle strength. Caspase and calpain initially disrupt the structure of actin and myosin myofibrils, which reduce contractility with the early loss in force generation [13,27]. These proteases disrupt myofilament structure by cleaving actin and myosin from sarcomeres, and supply proteins to the UPS, which is activated by pro-inflammatory cytokines. Tumor necrosis factor (TNF) and NFκB accelerate ubiquitination: the process of marking proteins for degradation. 

Caspases are active not only in the initiation of protein breakdown by UPS, but also in apoptosis (caspase3). Calpains are calcium-dependent cysteine proteases that are involved in the function of the UPS (m and μ calpains), with calpain3 (also known as p94) being specific to muscle. It plays an important role in activated protein degradation during sepsis. It is possible that disruption of myofilament structure contributes to the early loss in force generation [13]. 

Sarcopenia largely affects fast-twitch muscle fibers. Reduction in muscle mass and strength leads to impaired mobility, along with an increased risk of falls, compromised independence, and lowered quality of life [6].

### 3.2. Decrease in Calcium Release from the Sarcoplasmic Reticulum

The function of calpains is calcium-dependent, as is the activation of ubiquitination. Thus, calcium regulates protein breakdown. Above all, it is essential for muscle contraction, has a direct effect on myosin ATPase, and influences glycolysis and oxidative metabolism. Calcium homeostasis is ensured in the sarcoplasmic reticulum. Outflow of Ca^2+^ through ryanodine receptors is ATP-dependent. ATP deficiency reduces Ca^2+^ release from the sarcoplasmic reticulum, affecting membrane excitability of skeletal muscle. Force generation is, thus, rapidly reduced during sepsis [4,27,28,29]. 

### 3.3. Sodium Channels and Electrical Inexcitability

Excitation is necessary to generate an action potential, which leads to changes in the permeability of the membrane to sodium, potassium, and possibly calcium ions. Following excitation, the resting energy potential (−80 to −90 mV) increases to reach the threshold potential at which sodium channels open. Na^+^ thus enters the intracellular space, and the inner side of the membrane becomes increasingly positively charged compared to the outer side of the membrane, resulting in depolarization (+20 to +30 mV). Potassium channels then open and K^+^ leaks out of the cell to maintain electroneutrality, and membrane voltage decreases, causing repolarization. In septic patients, alterations to membrane sodium pumps can cause disturbances in electrical excitability. Inflammatory cytokines have a neurotoxic effect, causing chronic membrane depolarization, functionally manifesting as “denervation”. The resting membrane potential of muscle fibers is reduced and is unable to reach the action potential, as voltage-gated ion channels follow an all-or-nothing rule. Thus, the rapid development of muscle weakness is initially more of a functional issue, affecting both nerves and muscles [13,30,31].

## 4. Mitochondrial Dysfunction

### 4.1. Muscle Mass, and Polyneuromyopathy in the Critically Ill

Sufficient energy is a prerequisite for muscle contraction; therefore, mitochondria, the organelles that are the “power plant” of the organism, are central to muscle function.

Mitochondria are subcellular organelles that furnish the cell with adenosine triphosphate (ATP), which they generate by oxidative phosphorylation, and are absolutely essential for the energetic processes of every cell. In addition to this extremely important function, they are involved in calcium homeostasis and intracellular reactive oxygen species (ROS) generation, mediate intracellular communication, and regulate apoptosis. Mitochondrial damage is associated with a lack of released energy (ATP), ROS overproduction, and the release of cytochrome c. Ultrastructural damage to the mitochondria and mitochondrial dysfunction also contribute to organ failure [32,33]. 

Mitochondria are sensitive to oxidative damage. On the one hand, mitochondria are the main source of free radicals, and on the other hand, are a key target for oxidative damage. Oxidative stress is typical during sepsis, a dysregulated response to severe infection, but it is also quite common during aging. Senescence is accompanied with a progressively increasing accumulation of ROS in line with the gradual decline in the capacity of the antioxidant defense system. When the defense system is no longer able to cope with the enhanced rate of oxidant production, cellular and subcellular environments become more susceptible to damage. Mitochondrial dysfunction has been demonstrated to activate cell apoptosis and can ultimately result in organ damage [34,35]. 

Mitochondrial dysfunction is, thus, a key player in the development of ICUAW in the context of critical illness, most progressively in sepsis. Age-related mitochondrial dysfunction plays an equally important role in the development of sarcopenia with the progressive decline in mitochondrial bioenergetics. The typical manifestation is a reduction in maximal oxygen uptake (VO_2_ max) and a consequent decrease in exercise tolerance [33,36]. 

Further, alteration of mitochondrial permeability transition pores (mPTP) is one of the mechanisms of sarcopenia, and results in ROS overproduction, and triggers muscle atrophy by activating the FoXO transcription factor family and ubiquitin ligases, thus activating proteolysis. Open mPTP can lead to the release of pro-apoptotic factors and released cytochrome c can increase proteasomal activity [37].

### 4.2. Mitochondria in Skeletal Muscle

Skeletal muscle enables movement and maintains posture, but it also has a role in thermoregulation, ensuring nutritional balance, glucose uptake, and is a hormone source [38,39]. Myokines are released from muscle during exercise; they are proteins with endocrine and paracrine functions that control inflammatory processes, angiogenesis, and myofibril hypertrophy, and also modulate fuel oxidation [38]. 

Skeletal muscle is a highly energetic tissue that typically contains mitochondria in three locations: subsarcolemmal, perinuclear, and intermyofibrillar. While subsarcolemmal mitochondria provide resistance to ROS, intermyofibrillar mitochondria are the source of ATP during oxidative phosphorylation and Ca^2+^ modulation. With regard to metabolism, the energy source in red myofibers (type I) is slow oxidative (SO) aerobic oxidation. Fast (or white) muscle fibers use glycolysis as the energy source, with fast oxidative and glycolytic reactions (FOG) in type IIa, and only fast glycolytic (FG) reactions in type IIb fibers [33]. 

#### 4.2.1. The Warburg Effect

Hyperglycemia due to stress metabolism is typical of conditions such as sepsis, SIRS, and traumatic brain injury with the presence of pro-inflammatory cytokines. The latter with hyperlactatemia is a marker of unfavorable prognosis. New findings suggest that hyperglycemia is affected by markedly upregulated gluconeogenesis and insulin resistance (with blocked GLUT4 transporters preventing glucose entry into muscle), as well as by persistent glycolysis even in the presence of the adequate oxygen substitution, the so-called Warburg effect. In macrophages and dendritic cells, lipopolysaccharide (LPS) activates inducible NO synthase (iNOS), which increases the production of nitric oxide (NO), which in turn nitrosylates iron–sulfur proteins in the mitochondrial electron transport chain, leading to inhibition of oxidative phosphorylation. LPS also stimulates mTOR, and increases production of hypoxia-inducible factor 1α (HIF-1α) that, in turn, can inhibit TCA (Krebs cycle), thus potentiating oxidative glycolysis [40]. 

#### 4.2.2. Lactate Shuttle

Lactate is not only an end-product of glycolysis, as was thought until recently, but it is also an important metabolic substrate. Lactate has an important role as an energy source, and also acts as a signaling molecule for metabolic regulation. Skeletal muscle plays an indispensable role in lactate shuttling, i.e., exchanging this energy source between the tissues or cells that produce it and those that utilize it. A typical example is during physical exercise, when skeletal muscle produces lactate for the myocardium, brain, or liver (Cori cycle, metabolization of lactate back to glucose). However, the lactate shuttle also has a role in cell–cell transport, between white muscle fibers that produce it, and red muscle fibers that consume it. Thus, lactate appears to be the major fuel for red skeletal muscle, the heart, and the brain under conditions of increased energy demand [41]. 

### 4.3. Mitochondrial Quality Control Mechanisms

Mitochondria have to adapt rapidly to changing energy requirements, energy supply from aerobic or anaerobic pathways, and oxidative stress. This plasticity of mitochondria in response to energy demand is regulated by multiple molecular signals. This regulation happens in stages ranging from mitochondrial biogenesis, through mitochondrial dynamics (fusion and fission), to mitochondrial autophagy (mitophagy). A reduced capacity for mitochondrial quality control leads to mitochondrial dysfunction—a key marker of multiple organ failure. Skeletal muscles are among those that are affected early and significantly [32].

#### 4.3.1. Mitochondrial Biogenesis

Mitochondrial biogenesis is the process by which new mitochondria are produced. The main regulator of mitochondrial biogenesis is proliferator-activated receptor γ coactivator 1α (PGC-1α). This interacts with other transcription factors such as FoXO, hepatocyte nuclear factor (HNF 4α), or nuclear respiratory factor (NRF1, NRF2), and activates transcription from replication of MtDNA (Figure 4).

#### 4.3.2. Mitochondrial Dynamics

Mitochondria adapt their number, morphology, and size to the energy requirements and can either fuse or divide by fission under the influence of GTPase. GTPases controlling mitochondrial fission include the fission protein dynamin-related protein1 (Drp1) and its receptor mitochondrial fission protein (Fis1), the mitochondrial fission factor (Mff), and mitochondrial dynamics proteins (MiD49, MiD51). Fusion is controlled by mitofusin (Mfn) and optic atrophy 1 (OPA 1) (Figure 5) [37,42,43,44,45].

#### 4.3.3. Mitochondrial Autophagy

The final step in quality control is mitochondrial autophagy or mitophagy. This process eliminates damaged organelles by forming a double-membrane autophagosome, which then fuses with the lysosome where the contents are degraded. This requires receptors that recognize damaged mitochondria, where the main regulator of mitophagy is Pink1/Park2. Pink1 (PTEN-induced putative protein kinase) and Park2 act as an E3 ubiquitin ligase. In healthy mitochondria, Pink1 moves from its position in the outer membrane to the inner mitochondrial membrane and when its N-terminal mitochondrial targeting sequence (MTS) is exposed to the matrix, it is recognized by matrix processing peptidase (MPP) that cleaves the p64 form of Pink1 to its p53 form, which is subsequently released into the cytosol and cleaved by the proteasome. Thus, Pink1 does not activate autophagy in healthy mitochondria. 

This is not the case in damaged mitochondria, where sepsis results in the loss of mitochondrial membrane potential. Pink1 does not move to the inner membrane but is sequestered in the outer membrane of the mitochondria, and subsequently recognized and labeled by the bound ubiquitin ligase Park2. Mfn (mitochondrial fusion protein) is also labeled and subsequently cleaved by the proteasome, preventing mitochondrial fusion. Park2 builds ubiquitin chains that can be sufficient to recruit autophagic receptors such as protein 62 (p62), nuclear dot protein (NDP52), optineurin, and others to initiate autophagosome formation, thus triggering mitophagy. This allows damaged mitochondria to be removed. Inhibition of this “scavenging” action leads to the progression of MODS, thereby increasing mortality. The accumulation of damaged mitochondria has shown to trigger motor neuron and muscle fiber degeneration, and upon progression of muscle dysfunction, can lead to ICUAW or sarcopenia [32,46,47,48] (Figure 6).

## 5. Potential Targets for Intervention in ICUAW

There are no interventions that can consistently treat ICUAW or sarcopenia; therefore, most interventions focus on reducing or eliminating risk factors. Nonetheless, several new potential targets of signaling pathways are currently being evaluated in clinical trials.

### 5.1. Mitochondrial Monitoring and Therapy

Monitoring mitochondrial dysfunction is already possible using PCR on mitochondrial DNA (MtDNA). A new highly sensitive, quantitative droplet method can monitor MtDNA copies in stimulated peripheral blood mononuclear cells. Reduction in their levels is a sign of oxidative stress both in critical illness and in aging, where they are a sign of frailty [33,49]. 

Oxidative stress (ROS) and nitric oxide (NO) inhibit electron transport chains and leads to mitochondrial swelling. The current approach is to modulate action at the subcellular level. Activation of mitochondrial biogenesis is used for mitochondrial therapy; in sepsis, this effort to recover mitochondrial function can prevent organ failure. L-carnitine, succinate, ATP-MgCl_2_, cytochrome c, and ubiquinol (CoQ) all have the same effect. Inhaled CO can rescue liver failure in sepsis (by inducing heme oxygenase-1-mediated NF-E2-related factor 2) and CoQ can prevent LPS-induced mitochondrial dysfunction by improving mitochondrial biogenesis [33,50,51]. Bezafibrate, a drug used for dyslipidemia, is an agonist of peroxisome proliferator-activated receptor (PPAR) and can increase PGC-1α expression [52]. Similarly, the oral antidiabetic drug metformin activates mitochondrial biogenesis. Glutathione and melatonin can be targeted as mitochondrial antioxidants [33]. 

LPS disrupts mitochondrial physiology in skeletal muscle via its pleiotropic effects on sphingolipid metabolism. Treatment with myocrin, a de novo sphingolipid biosynthesis inhibitor, ameliorates skeletal muscle dysfunction by decreasing sepsis-induced Drp1 expression and subsequently reverting mitochondrial morphology, inhibiting excessive mitochondrial fission and restoring the balance between fusion and fission [46]. However, therapeutic intervention on mitochondrial dynamics aimed at suppressing muscle atrophy and the development of sarcopenia—e.g., by suppressing Drp1 overexpression to attenuate the aging-related accumulation of mitochondrial dysfunction and sarcopenia—can have conflicting results. The network of signaling cascades is very dense with many feedback loops. Mitochondrial fission must be maintained to ensure mitochondrial and muscle health [53]. 

Impaired mitophagy also contributes to amplifying organ failure in sepsis. Targeting this scavenging process is another option for mitochondrial therapy. Carbamazepine, lithium, and sodium valproate are used as autophagic flux enhancers. Rapamycin and activated protein C are pharmacological agents used to induce autophagy. However, rapamycin affects many other metabolic pathways (e.g., proteosynthesis through the mTOR system) [54]. Current nutritional recommendations for critically ill patients are based on this principle. Hypocaloric nutrition with a gradual increase in energy and protein is recommended at the beginning so that autophagy does not become “excessive” [49]. Adenosine monophosphate-activated protein kinase (AMPK) and the silent mating-type information regulation 2 homolog sirtuin (SIRT1) are two of the best-known metabolic sensors that can directly affect PGC-1α sensitivity and mitochondrial biogenesis [54]. 

Moderate long-term exercise stimulates metabolic adaptations in aged skeletal muscle through the activation of PGC-1α, AMPK, and the SIRT1 pathway. Endurance exercise not only stimulates mitochondrial biogenesis but also causes repeated oxygen free radical loading. This results in an increase in antioxidant capacity. In addition to ROS production during exercise, myofibrils are damaged and, thus, proteolysis is stimulated after exercise with an increase in the amino acid pool and subsequent stimulation of proteosynthesis, thus increasing muscle mass and strength. Regular exercise tends to maintain low levels of oxidative damage and improve proteostasis, preventing sarcopenia [33,55,56]. Blood flow-restricted exercise (BFR) using a pneumatic tourniquet system in working musculature results in inadequate oxygen supply. BRF mediates muscle hypertrophy through protein signaling and satellite cell proliferation to a greater extent than resistance exercise alone [4,57]. The critically ill and often geriatric sarcopenic patients are unable to exert sufficient training loads to increase muscle mass. Neuromuscular electrical stimulation shows positive results for preventing skeletal muscle weakness and wasting in critically ill patients [58].

### 5.2. The Critical Nature of the Proteolytic System

Muscle is metabolically very active and is constantly challenged by mechanical, oxidative, and heat stresses. A properly functioning proteolytic system is essential to maintain proper function and tissue recovery. Improper proteolysis leads to myopathies; given this picture of the negative impact of proteolysis, its original objective of adaptation to stress is lost. Activated skeletal muscle proteolysis and its effect on the development of polyneuromyopathy in critically ill patients can be used in efforts to suppress proteolysis, but proper timing of proteasome inhibition is therefore crucial [14].

Skeletal muscle cell (myoblast) differentiation depends on the early activation of appropriate myogenic factors. UPS is involved in cell differentiation, initially removing the paired-box transcription factors Pax3 and Pax7, thereby initiating the transformation of satellite cells. Its role is critical for the early activation of the key myogenic factor MyoD by removing its inhibitor Id, and other myogenic factors requiring proteolytic cleavage such as E2A proteins, filaminB, Myf5, and myogenin. This initiates the transformation of the satellite cells first to myoblasts and then to myotubes. All the ROS generated during the entire process of myogenesis also need to be scavenged [59,60]. 

During muscle hypertrophy following resistance exercise, both proteosynthesis and degradation of damaged proteins are increased, and UPS (through the proteasome axis (MuRF1—muscle-specific RING finger protein1, MAFbx—atrogin1)) facilitates myofilament restructuring and growth [61]. 

Caspases, besides initiating apoptosis, have an important role in skeletal myoblast differentiation. They activate satellite cells following the cleavage of the promyogenic kinases MST1, HIPK2, NEK5, and CAD and the cleavage of Pax7 (caspase3). Caspase3 enhances myoblast fusion, a critical step in muscle maturation [59,62]. As it is one of the major protein degradation pathways, exacerbated autophagy can induce ICUAW. On the other hand, “basal” autophagy is necessary to maintain muscle mass and prevent muscle atrophy. It has an important role in preventing sarcopenia, where stem cell senescence limits muscle regeneration in aging. Without autophagy, efficient clearance of damaged organelles and proteins would not work during muscle stress adaptation following exercise or starvation. Exercise can activate beclin1 (by phosphorylation and release from the BCL2–beclin1 complex), allowing autophagosome formation. The extent of activation of autophagic signaling is influenced by the duration and intensity of exercise. Autophagy is activated during exercise in ultra-endurance runners; the highly increased skeletal muscle expression of key autophagy genes such as Atg4b, Atg12, LC3, Bnip3, etc., is necessary for the removal of dysfunctional proteins and to meet the elevated energetic demands. FOXO3 phosphorylation decreases at the same time as proteasome activation. So, balanced autophagy is necessary to maintain normal protein turnover [4,59] (Figure 7).

### 5.3. The Role of Starvation and Stress Metabolism

Nutritional supplementation plays an important role in the prevention of ICUAW and the development of sarcopenia, especially protein sufficiency, in an attempt to influence the link between proteosynthetic mTOR and proteolytic UPS. The main stimulators of mTOR include branched-chain amino acids, especially leucine and its metabolite hydroxymethylbutyrate (HMB) [63]. Leucine uses both insulin-dependent (IGF-IR/Akt/mTORC1) and insulin-independent (RagD/mTORC1) pathways to stimulate mTOR. This latter pathway of activation via Ragulator is common to both leucine and its metabolite HMB. Other amino acids use other activation pathways, e.g., arginine and glutamine use the lysosomal-sensing protein SLC38A9/mTORC1 [26]. 

An important role is also played by sufficient vitamin D, considered to be a steroid hormone. Vitamin D receptors (VDR) are found in various tissues, and are also presented on skeletal muscle, predominantly in fast-twitch muscles. Vitamin D is associated with oxidative stress, muscle energy metabolism, mitochondrial functions, and acts in Ca^2+^ homeostasis, which is necessary for muscle contraction. By reducing Ca^2+^ reuptake into the sarcoplasmic reticulum, it prolongs the relaxation phase of muscle contraction. Vitamin D deficiency decreases protein synthesis (through a signaling cascade: decreased IGF-1/Akt/mTOR), and on the other hand, decreased IGF-1/Akt activates FoXO and triggers muscle atrophy by elevation of MuRF1 and atrogin-1. UPS is also stimulated via attenuation of steroid receptor coactivator complex (Src) and decreased PGC-1α. The availability of vitamin D decreases with age, so it has a role in sarcopenia. Supplementation of vitamin D, to a minimum level of 75–100 nmol/L (30–40 ng/mL), reduces the symptoms of myopathy. Of course, these problems cannot be fully solved with vitamin D alone; what is important is bioavailability and the right level [64,65]. Nonetheless, robust evidence for this is still lacking.

### 5.4. The Role of Glucocorticoids

Systemic inflammation and elevation of cytokines (TNF, IL1, IL6) drive muscle atrophy. They activate the mitogen-activated protein kinase (MAPK) and upregulate atrogin-1 (MAFbx) and MuRF1. 

Glucocorticoids are commonly used in patients to suppress cytokine storms in critically ill patients, including COVID-19 patients [66]. However, they have a markedly negative impact on muscle dysfunction, activating UPS with the concomitant increase in atrogin-1 and MuRF1. Therapeutically, glucocorticoid receptor blockade can be used to manage glucocorticoid-induced skeletal muscle atrophy. This may attenuate UPS activated by acidosis, insulin resistance, or sepsis. There are numerous inhibitors to block MAFbx and/or MuRF1, including eIF3-f, MyoD, and myogenin. However, this therapy has had indifferent success, possibly because of the different causes of atrophy, multiple signaling pathways, and no one specific inhibitor for all stages of atrophy [22,23]. Nevertheless, IL1 blockade with anakinra has shown admirable results in survival of sepsis patients [67,68,69]. 

### 5.5. Specifics of Therapy for Sarcopenia

Therapies for sarcopenia, or efforts to slow muscle loss due to aging, rely on sufficient intake of quality protein (leucine, phenylalanine, and arginine—with new recommendations to increase protein intake up to 1.5 g/kg/day), limitation of saturated fats, and sufficient omega3 fatty acids and vitamin D, along with exercise. Active lifestyle and exercise can improve repair myogenesis and increase the expression of neuronal form of NOS in the muscle, thus leading to the activation of satellite cells (SC). SC are located between the basal lamina and sarcolemma of muscle fibers; they are the principal contributors to muscle repair and growth and decrease significantly during aging. Moreover, myostatin, an extracellular messenger of the transforming growth factor superfamily, inhibits factors that regulate myogenesis. Exercise has been observed to decrease this muscle growth-inhibiting messenger. During exercise, increased levels of vascular endothelial growth factor (VEGF) and epidermal growth factor (EGF) modulate oxidative stress, improve brain-derived neurotrophic factor (BDNF), and thus improve stimulation of skeletal muscle contraction [70,71,72,73,74]. 

Several clinical studies have explored possible pharmacological interventions in signaling pathways to manage sarcopenia. Among them are myostatin inhibitors (Bimagrumab and the more recent trevogrumab and domagrozumab). Ghrelin, important for food intake with anabolic properties, is being tested for appetite reduction and malnutrition. ACE inhibitors, such as perindopril or the selective AT inhibitor losartan, are also known to be beneficial by increasing IGF-1 levels [75].

## 6. Conclusions

During muscle catabolism, the rate of protein breakdown exceeds the rate of protein synthesis. 

Activated proteolysis is of primary importance in ICUAW. Two major pathways act to facilitate muscle degradation—UPS and dysregulated autophagy. Nutrition and rehabilitation are the major factors in treatment or prevention: early mobilization, exercise (in-bed cycle ergometry, blood flow-restricted exercise) and electromyostimulation (EMS), along with intake of branched-chain amino acids, bolus enteral nutrition, insulin, and glucose target therapy. New therapeutic options are being considered to influence mitochondrial dysfunction, systemic inflammation, and the signaling cascades involved.

Sarcopenia is the loss of muscle strength and muscle mass during senescence or secondary due to inactivity or malnutrition. The fundamental reason for this is the gradual decrease in proteosynthesis and, thus, a deficit in muscle regeneration. Chronic low-grade pro-inflammatory conditions, accompanied by compromised immune response during aging, known as “immune-senescence” or “inflammaging”, oxidative stress, ROS, and motor neuron loss all contribute to the condition. Again, nutrition rich in quality protein, an active lifestyle with exercise, antioxidants, inhibitors of myostatin, and other signaling cascades are crucial for prevention and therapy.

The etiology of both these muscle catabolic states is multifactorial, and hence, multifactorial approaches are needed to address them. Mitochondrial dysfunction is the link between ICUAW and sarcopenia, and thus figuratively between critical illness and senescence (aging). It appears that critical illness accelerates the aging process. Mitochondrial dysfunction appears to be the link also at the subcellular level in the pathophysiology of critical illness and senescence; that is, between the two scenarios of muscle impairment, sarcopenia, and polyneuromyopathy in the critically ill.

## Figures and Tables

**Figure 1 ijms-23-08396-f001:**
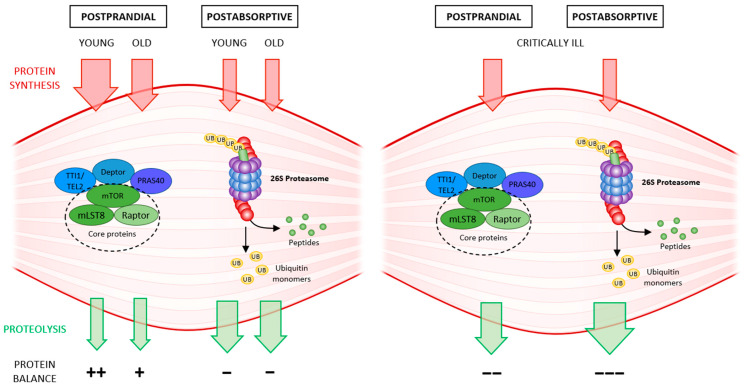
Protein turnover in the young, the old, and the critically ill partly according to [11]: In old patients, protein synthesis decreases postprandially due to anabolic resistance; compared to young patients, the protein balance becomes negative over time, leading to sarcopenia. Proteolysis between meals postabsorptively does not change significantly. In critically ill patients, on the one hand, proteosynthesis is affected due to anabolic resistance, but above all, proteolysis is markedly activated, for the need of protein as source of stress metabolism-activated gluconeogenesis. Protein balance is strongly negative; ICUAW develops rapidly.

**Figure 2 ijms-23-08396-f002:**
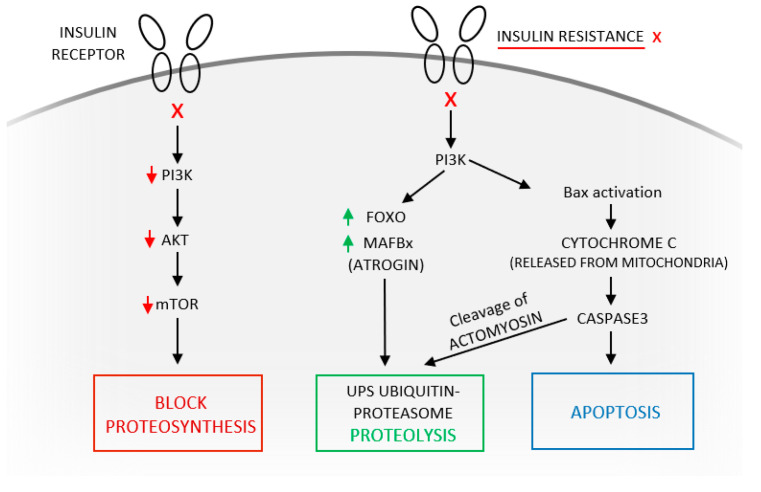
Signaling pathways that activate caspase3 and the UPS in skeletal muscle. Schematic of the main molecular pathways balancing muscle protein synthesis and proteolysis. Sepsis, inflammation, and immobility shift this balance towards protein breakdown. X means insulin resistance, block of insulin receptor.

**Figure 3 ijms-23-08396-f003:**
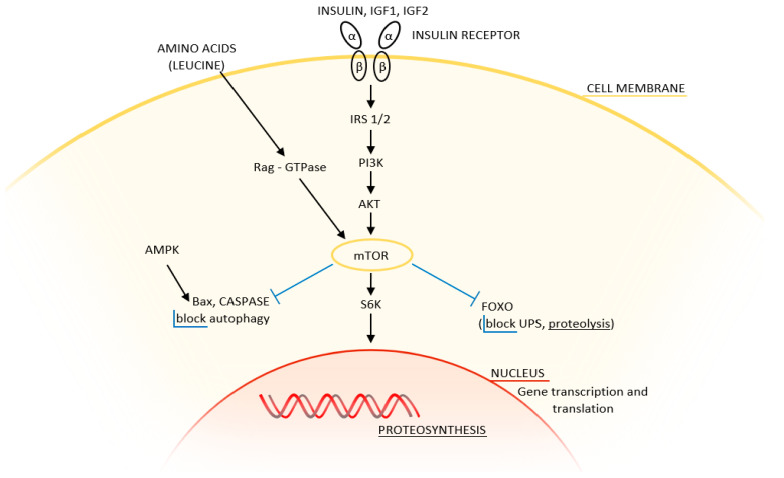
Signaling pathway of muscle protein synthesis. Insulin and insulin-like growth factors (IGF) act through phosphoinositol-3-kinase (PI3K) and Akt kinase to activate mammalian target of rapamycin (raptor mTORC1 and rictor mTORC2). Adequate supply of amino acids (leucine) in the diet activates Rag GTPase and activates raptor mTORC1, but not rictor mTORC2.

**Figure 4 ijms-23-08396-f004:**
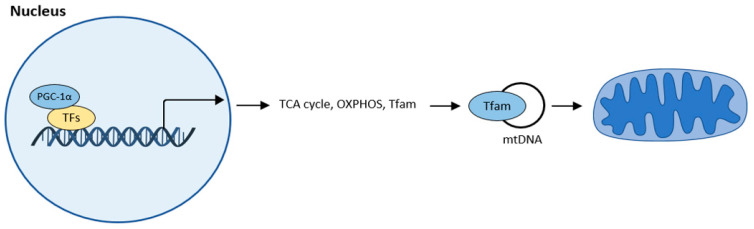
Mitochondrial biogenesis according to [32]. The central regulator of mitochondrial biogenesis: peroxisome proliferator-activated receptor γ coactivator 1α (PGC-1α). Transcription factors (TFs): forkhead box class-O (FoxO1), hepatocyte nuclear factor 4a (HNF4a), nuclear respiratory factor (NRF1, NRF2).

**Figure 5 ijms-23-08396-f005:**
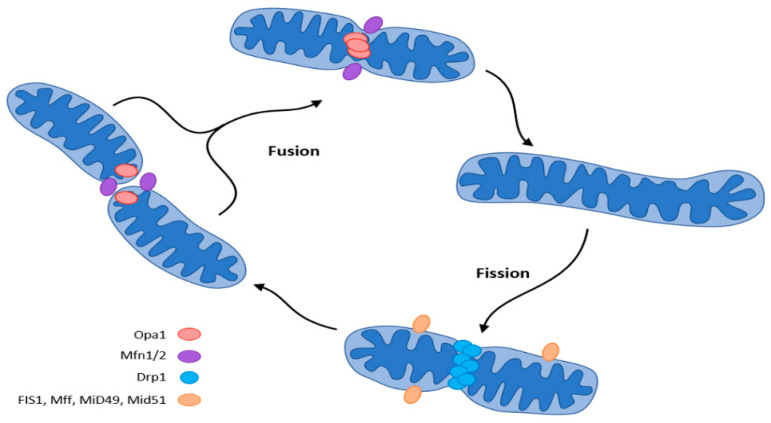
Mitochondrial dynamics (fission and fusion events) according to [37]. The central regulator of mitochondrial biogenesis: peroxisome proliferator-activated receptor γ coactivator 1α (PGC-1α). Transcription factors (TFs): forkhead box class-O (FoxO1), hepatocyte nuclear factor 4a (HNF4a), nuclear respiratory factor (NRF1, NRF2).

**Figure 6 ijms-23-08396-f006:**
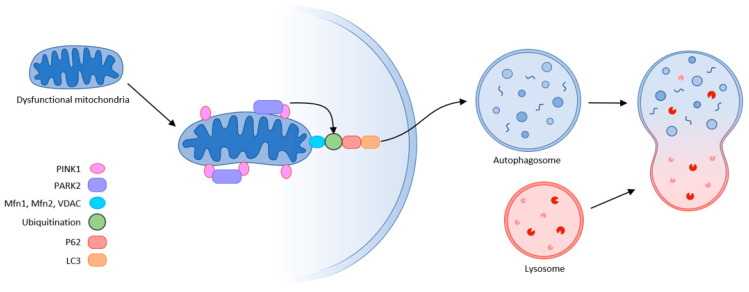
Mitophagy (schematic representation of the autophagy machinery) partly according to [32].

**Figure 7 ijms-23-08396-f007:**
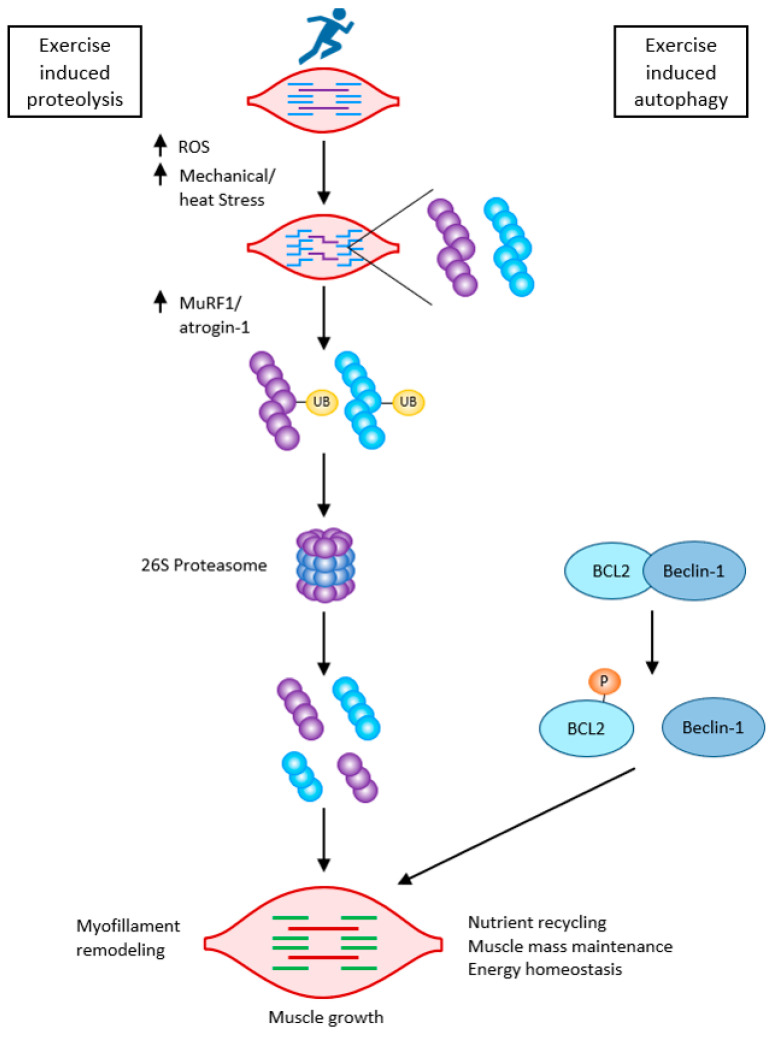
Exercise-induced muscle growth through induced proteolysis and induced autophagy according to [59]. Exercise-induced protein damage via increased ROS/mechanical and heat stress, increased MuRF1, and atrogin-1 (MAFbx), and both muscle-specific ubiquitin ligases lead to the activation of the 26 S proteasome to rid the cells of non-functional myofibrillar proteins. Exercise also activates autophagy: beclin-1 is phosphorylated and released from the BCL2–beclin-1 complex. Exercise-induced autophagy is necessary for the clearance of damaged organelles and proteins. This is critical for skeletal muscle remodeling and growth.

## Data Availability

Not applicable.

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
