# Peer review of "Molecular Mechanisms Underlying Intensive Care Unit-Acquired Weakness and Sarcopenia"

_ijms, 2022, doi:10.3390/ijms23158396_

Round 1

Reviewer 1 Report

For the author:

I thank to the editors for the opportunity to review this study. The present manuscript by Marcela Kanova and Pavel Kohout analysed “Molecular Mechanisms Underlying Intensive Care Acquired Weakness and Sarcopenia”. This review, it focuses to compare the molecular mechanisms underlying the two types of muscle atrophy, and address questions regarding possible shared molecular mechanisms, and whether critical illness accelerates the ageing process. First, before analysing the review in depth, I would like to clarify the following doubts. The authors have not clearly shown what problems exist and why this review is necessary and what are the motivations and stimuli that have generated the need to carry out this review. This question should be added in the main document. Indeed, after a quick search there are other reviews similar to their study, such as:

-          Acquired Muscle Weakness in the Surgical Intensive Care Unit: Nosology, Epidemiology, Diagnosis, and Prevention. (DOI: 10.1097/ALN.0000000000000874) (2016).

-          Intensive Care Unit-Acquired Weakness: A Review of Recent Progress with a Look Toward the Future. (DOI: doi: 10.3389/fmed.2020.559789) (2020).

-          Intensive Care Unit-AcquiredWeakness: Not Just Another Muscle Atrophying Condition. (DOI: 10.3390/ijms21217840) (2020).

-          Muscular weakness and muscle wasting in the critically ill. (DOI: 10.1002/jcsm.12620) (2020).

-          Sarcopenia in critically ill patients (DOI: 10.1007/s00540-016-2211-4) (2016).

-          Intensive care unit-acquired weakness and the COVID-19 pandemic: A clinical review (DOI: 10.1002/pmrj.12757) (2022).

1.     What does scientific evidence provide your review that these reviews do not offer?

2.     On the other hand, of the 67 references that the authors have (a rather deficient number for a review) 39 references are prior to 2018, I think we are talking about a current topic and that there are enough bibliographic sources to focus all references in the last 5 years.

 - What were the Strengths and limitations of this review?

Author Response

Response to Reviewer 1 Comments

Thank you for carefully reading our work and posing some stimulating questions.

The authors have not clearly shown what problems exist and why this review is necessary and what are the motivations and stimuli that have generated the need to carry out this review. This question should be added in the main document.

We explained this main idea why we wrote the review in the Cover Letter but did not emphasize it enough in the text. Thank you for the pointing our attention towards this issue. We have added our main motivation to the main text of the review.

ICUAW is primarily a clinical diagnosis. Our main motivation was to present a clinician/intensivist perspective on ICUAW. To state what is decisive for us in critically ill patients (with ARDS, septic shock, polytrauma, MODS) on long-term artificial ventilation and other types of organ support, including ECMO, often with the necessity of sedation (despite the efforts to reduce it and start with early rehabilitation), and other medication, such as corticosteroids, antibiotics, possibly also muscle relaxants.

We are well aware of the fact that despite the progress in intensive care medicine in improving the short-term outcome (28-day mortality), the long-term consequences, the quality of life of patients who have survived a critical illness, are still not good. Patients who have survived a critical illness suffer from a whole constellation of symptoms of Post Intensive Care Syndrome (PICS), of which polyneuromyopathy is an important part. Every clinician is certainly interested in what can be improved. And this is what we strive for in practice with the maximum nutritional and rehabilitation care provided for our patients not only in the ICU but we also invite long-term ventilated patients to the post-ICU clinic.

What scientific evidence does your review provide that these reviews do not offer?

Our review, thanks to clinical monitoring in the ICU and subsequently in the post-ICU outpatient clinic, emphasizes the common link between ICUAW and sarcopenia in a multifactorial aetiology, this being the level of cytokines (TNF, IL1, IL6), i.e., the severity of the critical illness for the development of ICUAW and chronic inflammation during aging (Inflammaging) and chronic diseases for the development of sarcopenia. This leads to mitochondrial dysfunction with problems in energy production, leading to excessive ROS with destroying functions. And thus, we answer the initial question we set out, whether and what is the connection between the two nosological units. Critical illness accelerates aging. And we believe that this clinical insight will be inspiring for molecular biologists.

We also emphasize the long-term outcome – Post Intensive Care Syndrome. Patients pay for surviving a critical illness a price in the form of muscle loss, memory impairment, more frequent delirium, and post-traumatic stress disorder. Patients overcome these consequences only with difficulty and in the long term. Both authors (ESPEN members) deal with these consequences, using intensive metabolic care in cooperation with rehabilitation care.

On the other hand, of the 67 references that the authors have (a rather deficient number for a review) 39 references are prior to 2018, I think we are talking about a current topic and that there are enough bibliographic sources to focus all references in the last 5 years.

We have selected "core" works for primary citations in our review, not interpretations of them. These are the works of molecular biologists, biomedical engineers and other clinicians, which perfectly describe the complex problems of ICUAW and sarcopenia, and which appealed to us. It is difficult to distinguish to what extent critical illness contributes to muscle weakness in a highly heterogeneous group of critically ill patients with different severity of the condition, different means of treatment and organ support, and how much is contributed by sarcopenia in the pre-illness period (either primary, associated with age, or particularly secondary, associated with malnutrition and chronic diseases).

We would like to thank you for the suggestions of other works, especially the Shefold's work; we have tried to incorporate the ideas of the more recent ones into our review. However, a direct comparison of the molecular mechanisms of ICUAW and sarcopenia is only marginally mentioned there. Another recommended work of Wenkang Wang and Hety Lad has already been included in our review.

We hope that our work summarizes these mechanisms from the point of view of a critically ill patient with organ support, in a clinically usable depth, and complements the previous works.

Once again, thank you for your advice and I hope we have managed to explain our goal sufficiently.

We believe that we have been able, with your insights, to put together a high-quality paper which offers an interesting overview of this topic that is relevant to both healthy people as well as to critically ill patients.

Sincerely,

Marcela Káňová, Pavel Kohout

Reviewer 2 Report

This is a manuscript entitled “Molecular Mechanisms Underlying Intensive Care Acquired 2

Weakness and Sarcopenia”

” by Marcela Kanova et al.

This review is interesting, however this reviewer has some concerns below.

1.

Fig 2 is not easy to understand. The authors should show the meaning of the code. What means of “X”

2.

The title of Figure 5 and 6 should be changed

Author Response

Response to Reviewer 2 Comments

We would like to thank you for your comment. We really appreciate it.

This review is interesting; however, this reviewer has some concerns below.

1.

Fig 2 is not easy to understand. The authors should show the meaning of the code. What means of “X”

Thank you, especially for pointing our attention to the Figures. In Fig. 2, the “X” means insulin resistance, block of insulin receptor, as shown in Figure. We have added it to the description of the Figure for better understanding and clarification.

2.

The title of Figure 5 and 6 should be changed

We have changed the titles

Fig 5 Mitochondrial dynamics (fission and fusion events)

Fig 6 Mitophagy (schematic representation of the autophagy machinery)

We believe that we have been able, with your insights, to put together a high-quality paper which offers an interesting overview of this topic that is relevant to both healthy people as well as to critically ill patients.

Sincerely,

Marcela Káňová, Pavel Kohout

Reviewer 3 Report

The authors describe in detail the various aspects of intensive care-acquired weakness and sarcopenia. They explain molecular mechanisms of various metabolic pathways that contribute to this condition. The review is comprehensive and I recommend publication.

One minor suggestion about the mitochondrial dynamics section is to cite better papers. Currently, the citations on this topic do not seem diverse (Page 12, Fig.5).

For fission, well-known papers are:

1. Smirnova et al; Mol Biol Cell. 2001 Aug; 12(8): 2245–2256. Describes DRP1 is fundamental for mitochondrial fission.

2. Kalia et al; Nature volume 558pages 401–405 (2018). Gives a mechanistic understanding of the relationship of DRP1 with receptors (MiDs).

3. Otera et al; J Cell Biol (2010) 191 (6): 1141–1158. Describes MFF as a receptor for DRP1.

Similarly for mitochondrial fusion, Song et al; Mol Biol Cell. 2009 Aug 1; 20(15): 3525–3532.

Author Response

Response to Reviewer 3 Comments

We would like to thank you for your comment. We really appreciate it.

The authors describe in detail the various aspects of intensive care-acquired weakness and sarcopenia. They explain molecular mechanisms of various metabolic pathways that contribute to this condition. The review is comprehensive, and I recommend publication.

One minor suggestion about the mitochondrial dynamics section is to cite better papers. Currently, the citations on this topic do not seem diverse (Page 12, Fig.5).

For fission, well-known papers are:

  1. Smirnova et al; Mol Biol Cell. 2001 Aug; 12(8): 2245–2256. Describes DRP1 is fundamental for mitochondrial fission.
  2. Kalia et al; Naturevolume558, pages 401–405 (2018). Gives a mechanistic understanding of the relationship of DRP1 with receptors (MiDs).
  3. Otera et al; J Cell Biol(2010) 191 (6): 1141–1158. Describes MFF as a receptor for DRP1.

Similarly for mitochondrial fusion, Song et al; Mol Biol Cell. 2009 Aug 1; 20(15): 3525–3532.

We have added these recommended publications to the References.

We believe that we have been able, with your insights, to put together a high-quality paper which offers an interesting overview of this topic that is relevant to both healthy people as well as to critically ill patients.

Sincerely,

Marcela Káňová, Pavel Kohout

Round 2

Reviewer 1 Report

For the author:

I appreciate authors’ effort. The authors have obviously spent considerable time revising the manuscript and their hard work is clearly paying off. This manuscript is drastically improved from the original submission. The message is very clear, the language is much more clean, and the issues in the first version were corrected. Besides, the authors have answered all my comments successfully. For this reason, I encourage to editor to consider this manuscript for publication for the interesting value of the study realized, that now it is a much more robust study.

Reviewer 2 Report

The authors replied my concerns properly